# Light Energy Efficiency in Lettuce Crop: Structural Indoor Designs Simulation

**DOI:** 10.3390/plants12193456

**Published:** 2023-09-30

**Authors:** Luisa F. Lozano-Castellanos, Luis Manuel Navas-Gracia, Adriana Correa-Guimaraes

**Affiliations:** 1TADRUS Research Group, Department of Agricultural and Forestry Engineering, ETSIIAA, University of Valladolid, 34004 Palencia, Spain; adriana.correa@uva.es; 2Research Group on Biodiversity and Dynamics of Tropical Ecosystems—GIBDET, Faculty of Engineering Forestry, University of Tolima, Ibagué 730006, Colombia

**Keywords:** electrical consumption, indoor agricultural designs, indoor crops, lighting efficiency

## Abstract

Indoor agricultural offers efficient alternatives for intensive food production through automation technologies and controlled environments. Light plays a crucial role in plant development; however, photons captured by the crop are often wasted in empty spaces, resulting in low light efficiency and high energy costs. This research aims to simulate eight structural designs for an indoor lettuce crop, exploring different planting systems and light and culture bed combinations (static and mobile) to identify the most effective mechanism for light efficiency during crop growth. The simulations were carried out with spreadsheets based on applying formulas of yield in dry biomass per photosynthetic photons, lighting costs, harvest, and production. The results indicate that Circular Moving Light and Mobile Culture Bed with Quincunx Planting (CML-QM) and Circular Moving Light and Mobile Culture Bed with Linear Planting (CML-LPM) exhibit higher photon capture percentages (85% and 80%, respectively) and lower electricity consumption compared to static designs. The simulation results demonstrate the potential for significant improvements in photon capture and cost savings through optimized system designs. This investigation provides valuable insights for designing more efficient systems and reducing electricity consumption to enhance the capture of photosynthetic photons in indoor lettuce cultivation.

## 1. Introduction

Advances in lighting for indoor crops and horticulture have allowed controlled environmental parameters in closed environments and under different methods, handling and monitoring the plant development of a cultivar to guarantee its economic and ecological viability. Artificial light is one of the most important environmental parameters for developing and growing plants in indoor production systems. However, it has been strengthened by the replacement of conventional lamp technologies with Light-emitting Diodes (LEDs) [1] and by the use of technology and communication tools for the design of more efficient structures and adaptability to the species, production needs, available space, and cultivation methods that meet high-quality conditions [2]. Its efficiency must continue to be improved not only from the point of view of operational and production costs, but also from the real photon capture by the plants [1,3,4].

The proportion of irradiation emitted by the LED lamps is not significantly captured by the plants since most of it is reflected in the empty cultivation spaces or plants that grow during cultivation [3]. Minimizing these photon losses requires the analysis, forecasting, and control of the components of the light environment: photosynthetic photon flux density (PPFD or light intensity), the spatial arrangement, quality, cycle, and angle of the LEDs, of the characteristics of the cultivation space and the vegetative area (leaf area, growth, etc.) [5,6], especially in immobile systems that depend 100% on electronic devices for their operation.

There are various methods and strategies to improve lighting efficiency, e.g., evaluation of the grams of biomass produced and the photosynthetically active radiation (PAR) absorbed by plants, also known as light use efficiency (LUE); LUE quantification with morphological measures such as leaf area index (LAI) or projected canopy size (PCS) [3,7]; lamps with a high luminous efficacy or conversion efficiency into PAR; evaluation of the ratio of PAR received at the plant canopy by the PAR emitted from the lamps; PPFD distribution histogram estimated by the reflection of the canopy, exponential models of relative growth rate or leaf area growth rate, 2D and 3D modeling for vegetative growth [3]; LEDs with small viewing angle and the use of secondary optics [8], supplemental lighting with directional and mobility properties [9], spatial distribution simulation software of PPFD in cultivation space [6], among others.

In general, automation and control of elements in indoor systems are widely applied to increase lighting efficiency and agricultural productivity (e.g., [10,11,12]). However, its application in the development of crops in mobile beds is less documented, focusing on business experiences and cost–benefit analysis, and barely in the design, construction, and measurement of efficiency [13,14,15]; compared to the control of luminaries, which have a more extensive field in scientific documents [3,16,17,18], even though it has been established that the automatic or manual spacing of the plants throughout the cultivation period contributes to improving the LUE [19].

The present investigation highlights the importance of adequately taking advantage of the photonic capture available to plants through sensitivity analysis (simulation) in lighting systems with static and mobile lighting structural designs and crop bases with linear and quincunx planting systems. Lettuce was considered for this analysis. It is one of the most cultivated vegetable crops worldwide and the most stable and constant concerning production and harvested area [20]. Spain is one of the largest producers in Europe [21], and although indoor lettuce production is still a relatively small industry, it has been growing in recent years.

The main aim of this investigation is to identify the design of a realistic system that improves the total proportion of photosynthetic photons captured during indoor lettuce crop growth.

## 2. Results

The simulation of the proportion of the total photosynthetic photons captured by the leaves during the lettuce cycle under the design of Linear static light and Static Culture Bed with Linear Planting (LSL-LPS) and luminaires with 140° beam aperture is 50% (Figure 1). This structure would require an annual electricity consumption of approximately 4100 thousand kWh for a yearly production of 500 thousand kg and up to 62,000 thousand kWh for producing 7500 thousand kg of lettuce, representing an electricity annual cost of 955 and 14,325 thousand euros, respectively.

The Linear Static Light and Static Culture Bed with Quincunx Planting (LSL-QS) simulation and luminaires with a 140° beam opening is 55% (Figure 2). This structure would require an annual electricity consumption of 3775 thousand kWh for a yearly production of 500 thousand kg and up to 56,619 thousand kWh for the production of 7500 thousand kg, representing an electricity annual cost of 868 and 13,022 thousand euros, respectively. The additional 5% of photon capture in this system decreases the price per kilogram of lettuce to EUR 1.74, a difference concerning the base value of the LSL-LPS structure of EUR 0.17.

The Circular Moving Light and Static Culture Bed with Linear Planting (CML-LPS) (Figure 3) and the Circular Moving Light and Static Culture Bed with Quincunx Planting (CML-QS) (Figure 4) designs with 45, 90, and 140° aperture beam luminaires represent a photon capture of 60 and 65% during the lettuce cycle. They require an approximate annual electricity consumption of 3400 and 3100 thousand kWh, respectively, for producing 500 thousand kg and up to 51,000 and 47,000 thousand kWh for producing 7500 thousand kg. The annual cost of electricity for CML-LPS is 17% lower than LSL-LPS, achieving a price per kg of lettuce of EUR 1.59; and for CML-QS, it represents 23% less with a cost of EUR 1.47 per kg.

As for the Circular Moving Light and Mobile Culture Bed with Linear Planting (CML-LPM) (Figure 5) and Circular Moving Light and Mobile Culture Bed with Quincunx Planting (CML-QM) (Figure 6), both with an opening beam of 45, 90 and 140°. The first, with a capture of 80%, requires an annual electricity consumption of 2595 thousand kWh to produce 500 thousand kg of lettuce at a yearly cost of 597 thousand euros, and 38,925 thousand kWh to produce 7500 thousand kg with an annual cost of electricity of 8953 euros. On the other hand, CML-QM, with 85% capture, reduces annual electricity consumption to 2442 and 36,636 thousand kWh to produce 500 and 7500 thousand kg, achieving annual cost savings of 35 and 527 thousand euros compared to the design CML-LPM, and savings of 393 and 5898 thousand euros compared to the LSL-LPS design. With these designs, the cost per kg of lettuce is 1.19 and EUR 1.12.

Finally, the Linear Moving Light and Mobile Culture Bed with Linear Planting (LML-LPM) (Figure 7) and the Linear Moving Light and Mobile Culture Bed with Quincunx Planting (LML-QM) (Figure 8) have a photon capture percentage of 70 and 75, respectively.

For the production of 500 thousand kg of lettuce under the LML-LPM structure, there will be an annual electricity consumption of 2966 thousand kWh, compared to LML-QM, which would require 2768. The above represents a yearly electricity expense of 682 and 637 thousand euros. Compared with producing 7500 thousand kg of lettuce, 44,486 thousand kWh per year would be required with an electrical energy cost of 10,232 thousand euros using the LML-LPM structure. In contrast, with the LML-QM, 41,521 thousand kWh are needed per year with an expense of 9550 thousand euros per year. The cost to produce a kg of lettuce is EUR 1.36 for LML-LPM and EUR 1.27 for LML-QM.

In addition to the simulations on the percentage of photon capture achieved by each structural design, the annual electrical consumption (Table 1) and the annual electrical energy expenses (Table 2) are included below on hypothetical structures that achieve 90 and 95% use of light, and on the yearly production between 500 and 7500 thousand kilos of lettuce.

## 3. Discussion

Lettuce is an important crop for the Spanish economy, as it is one of the country’s most widely cultivated and consumed vegetables. Although there are no exact costs of lettuce production in Spain, the prices published by the Ministry of Agriculture, Fisheries and Food provide the approximate value for this crop, being EUR 0.91 per kilogram for week 13 of 2023 [22]. This price is lower than the cost per kilogram of lettuce under a hypothetical structure of 95% photon capture (EUR 1.01) and the simulated indoor CML-QM structure with 85% photon capture (EUR 1.12), which did not consider additional costs of production, such as the packaging and distribution of the product.

The cost of production of indoor lettuce can vary depending on the specific production system and technology used. However, indoor lettuce production tends to be more expensive than conventional production due to the higher energy and equipment costs [23,24]. Simulation shows that growing lettuce indoors can indeed be more costly but offer benefits such as year-round production, higher yields, lower water use, and investment recovery in the following years, depending on the system [25,26,27].

Searching for cost-reduction measures with interior mobile structures that improve energy systems and energy efficiency in plants is still under development. In typical systems in indoor agriculture, approximately 50% of the photon capture is achieved [28]. Structures such as those proposed in this research provide alternatives for improvement, especially during the initial phase of plant growth, where the space occupied by plant material in the culture bed is minimal, and the space occupied by irradiated light is more extensive [5,29,30].

The waste of light and the high electricity costs are controlled, among other alternatives, by the angle of the light beam and the on/off programming to define or limit the irradiated area to the size of the crop in its different phases [30,31]. This situation is reflected in CML-LPS and CML-QS, which achieve 60% and 65% photon capture by applying a 45° beam in the first phase of growth, 90° in the second and third, and 140° in the last with interleaved ignition lights. These structures, mainly due to the circular design of the luminaire [32], show savings in energy costs compared to LSL-LPS and LSL-QS, which maintain static lights with the same opening angle throughout the lettuce growth.

Modular and mobile indoor growing systems have been developed to maximize crop production in a limited space and minimize costs. These include levels with moving conveyor belts and pipelines that transport plants from one level to the next and employ closed systems and automated operating protocols to plant, cultivate, and harvest, allowing access and maintenance of crops through the mobility of the cultivation system [13,15,33,34,35].

The accordion-shaped growing bed design represents a form of automation that utilizes progressive movement during plant growth to optimize available space and favor photon capture and lettuce growth. While this design has demonstrated high performance, as in the LML-LPM and LML-QM structures, there are more options for further improvement.

Considering factors such as growth bed, light distribution, and plant density, exploring alternative approaches to maximize system efficiency and productivity is essential. As a result, there was increased photon capture, reduced light waste, and significant savings with the CML-LPM and CML-QM structures compared to other cultivation approaches. These findings support carefully considering the mentioned factors to maximize light-use efficiency in growing plants [36,37,38].

It is crucial to acknowledge the limitations and assumptions inherent in conducting simulations as an initial step towards characterizing electrical consumption and proposing cost optimization strategies based on photon capture percentages, contingent upon luminaire distribution and culture bed patterns. Firstly, this simulation was carried out with a specific plant species, which means that its applicability to other species may require consideration of varying crop yields, for example.

The use of a generic concept for luminaires and the reliance on recent electrical lighting costs in Spain, which can fluctuate throughout the year, introduce uncertainties in the proposed cost optimization strategies.

Future research should focus on conducting accurate tests of the simulations carried out to measure the photon capture and its veracity and other relevant aspects for the optimization of cultivation in controlled environments. Comprehensive studies on light characterization should be carried out, considering the influence of different light spectra and spectral distribution on plant growth and development. In addition, the effect of light color on plants’ physiological and metabolic processes and the quality of the final products should be investigated.

On the other hand, it is necessary to carry out economic and life cycle analyses to assess these farming systems’ financial viability and sustainability. This implies considering the costs of implementing and maintaining indoor structures, energy consumption, and environmental impacts. Likewise, developing new structures and technologies that maximize production and reduce energy expenditure should be encouraged, such as efficient lighting systems, water recirculation systems, and automated control systems. Together, these research directions will improve photon capture and move towards more efficient and sustainable cultivation systems in controlled environments.

## 4. Materials and Methods

Lettuce (*Lactuca sativa*) has been established as a prominent horticultural species in trials conducted in indoor agriculture [39,40]. Irrespective of the lettuce variety and the cultivation system employed, it is commonly planted at a standardized spacing of 20 cm between individual plants [41,42]. The growth cycle of lettuce typically spans approximately 48 days from the initial germination stage to the final harvest. In hydroponic systems, the dimensions of the lettuce plants can reach an approximate width of 25 cm and height of 24 cm, with slight variations depending on the species under cultivation [43].

### 4.1. Variables Simulation

The sensitivity analysis was carried out through simulations using Microsoft Excel 2016 spreadsheet based on the application of data, equations, and formulas of yield in dry biomass per photosynthetic photons (Equations (1) and (2)), lighting costs of systems with 100% contribution of artificial light (Equations (3)–(5)), harvest (Equations (6)–(10)) and production (Equations (11)–(13)). The analysis was developed in proposals for indoor structural designs (See Section 4.2) for the annual production of 500 to 7500 thousand kilos of lettuce during the four (4) most representative stages of growth.

#### 4.1.1. Yield in Dry Biomass by Photosynthetic Photons

The simulation begins by applying “The energy cascade model” (Equation (1)) for the potential yield in dry biomass of a crop in a controlled environment, followed by the identification of grams of dry biomass for each mole of photons (Equation (2)) [28,44,45].
(1)E=A×B×C×D
where:

E = Mol of carbon per mol photons;

A = Fraction absorbed photons;

B = Quantum yield;

C = Conversion efficiency on respiration;

D = Harvest index.
(2)F=E×G
where:

F = Grams of dry biomass per mole of photons;

E = Mol of carbon per mol photons;

G = Biomass per mol of carbon.

#### 4.1.2. Electricity Cost

The photonic efficiency of the LED is carried out as follows (Equation (3)):(3)$$H=\frac{I}{J}$$
where:

H = Photon efficiency;

I = Photon output;

J = Input electrical energy.

Then, the cost of electricity for each mole of photons is identified (Equation (4)):(4)$$K=\frac{L}{M}$$
where:

K = Cost of electricity per mole of photons;

L = Cost of electricity per kWh;

M = Performance (Equation (5)).
(5)$$M=\frac{\left(I\times 3600\times 10^{-6}\right)}{\left(\;J\times 10^{-3}\right)}$$

#### 4.1.3. Costs, Consumptions, and Mol of Photons per Kilogram

Cost per kg harvested (Equation (6)):

(6)N=L×Pwhere:

N = Cost per kg harvested;

L = Cost of electricity per kWh;

P = Electricity consumption per kg harvested (Equation (7)).

Electricity consumption per kg harvested (Equation (7)):

(7)P=Q×(1−R)
where:

P = Electricity consumption per kg harvested;

R = Water content of the species;

Q = Electricity consumption per kg of dry biomass (Equation (8)).

Electricity consumption per kg of dry biomass (Equation (8)):

(8)$$Q=\frac{1000}{F\times M}$$
where:

Q = Electricity consumption per kg of dry biomass;

F = Grams of dry biomass per mole of photons;

M = Performance.

Cost per kg of dry biomass (Equation (9)):

(9)S=L×Q
where:

S = Cost per kg of dry biomass;

L = Cost of electricity per kWh;

Q = Electricity consumption per kg of dry biomass.

Moles of photons per kg harvested (Equation (10)):

(10)$$T=\frac{1000\times \left(1-R\right)}{F}$$
where:

T = Moles of photons per kg harvested;

R = Water content of the species;

F = Grams of dry biomass per mole of photons.

#### 4.1.4. Cost of Electrical Energy Consumed for Lighting

The electrical energy consumed for lighting was calculated as follows (Equation (11)):(11)$$U=P\times V\times 10^{3}$$
where:

U = Electrical energy consumed for lighting;

P = Electricity consumption per kg harvested;

V = Annual production.

Then, the total annual cost is identified (Equation (12)):(12)W=L×U
where:

W = Total annual cost;

L = Cost of electricity per kWh;

U = Electrical energy consumed for lighting.

Finally, the cost per kilogram produced was calculated as follows (Equation (13)):(13)$$X=\frac{W}{V\times 10^{3}}$$
where:

X = Cost per kg produced;

W = Total annual cost;

V = Annual production.

### 4.2. Structural Designs

The simulations were developed in eight structural designs:Linear Static Light and Static Culture Bed with Linear Planting (LSL-LPS);Linear Static Light and Static Culture Bed with Quincunx Planting (LSL-QS);Circular Moving Light and Static Culture Bed with Linear Planting (CML-LPS);Circular Moving Light and Static Culture Bed with Quincunx Planting (CML-QS);Circular Moving Light and Mobile culture bed with linear planting (CML-LPM);Circular Moving Light and Mobile culture bed with quincunx planting (CML-QM);Linear Moving Light and Mobile Culture Bed with Linear Planting (LML-LPM);Linear Moving Light and Mobile culture bed with quincunx planting (LML-QM).

#### 4.2.1. Illumination

For this research, information from commercial luminaires was gathered to serve as a reference for the necessary characteristics in developing equations related to these luminaire features. These characteristics include a typical photon flux of 168 µmol/s and an input electrical energy ranging from 51 to 70 watts. These values pertain to general indoor agriculture luminaires and were essential for conducting simulations (Figure 9):One focused on linear static luminaires where during the productive cycles, will not have movement or the option of gradual ignition.Two luminaires with mobile structures or with progressive lighting from the inside to the outside (circular light) or from one end to the other (linear light) to increase the beam of light as the plant grows in width.

#### 4.2.2. Culture Bed and Planting Systems

Two static and mobile cultivation beds were considered for two planting systems: linear and quincunx. As a structural principle, the mobile bed will have a frontal displacement as an accordion (Figure 10). The dimensions of the growing beds are 10 cm wide, 80 cm long, and 20 cm growing distance with a distance of 50 cm from the lights.

## 5. Conclusions

This study analyzed various indoor system designs to identify the most efficient system for capturing photosynthetic photons during indoor lettuce crop growth. The simulation results revealed that structures with circular moving lights and mobile culture beds achieved the highest photon capture percentages, ranging from 60% to 85%. These systems showed the potential for reducing annual electricity consumption and lowering the cost per kilogram of lettuce compared to static lighting systems.

Static designs captured a lower percentage of photosynthetic photons. In contrast, mobile designs (bed and light) demonstrated even better results, capturing 80% to 85% of the photons and achieving significant cost savings, lighting efficiency, and profitability in lettuce production.

Efforts are underway to reduce costs and improve energy efficiency in indoor lettuce production systems. Automation, controlled lighting angles, and optimized space utilization are some strategies to minimize light waste and electricity costs. In addition, developing modular and mobile growing systems, such as accordion-shaped beds, presents new opportunities to improve efficiency and productivity.

Future research should focus on performing empirical tests to validate the simulations and explore other variables on plant growth and quality, considering implementation and maintenance costs, energy consumption, and environmental impacts. In addition, continued innovation in lighting systems, mobile beds, and automated control will contribute to more efficient and sustainable indoor growing practices.

## Figures and Tables

**Figure 1 plants-12-03456-f001:**
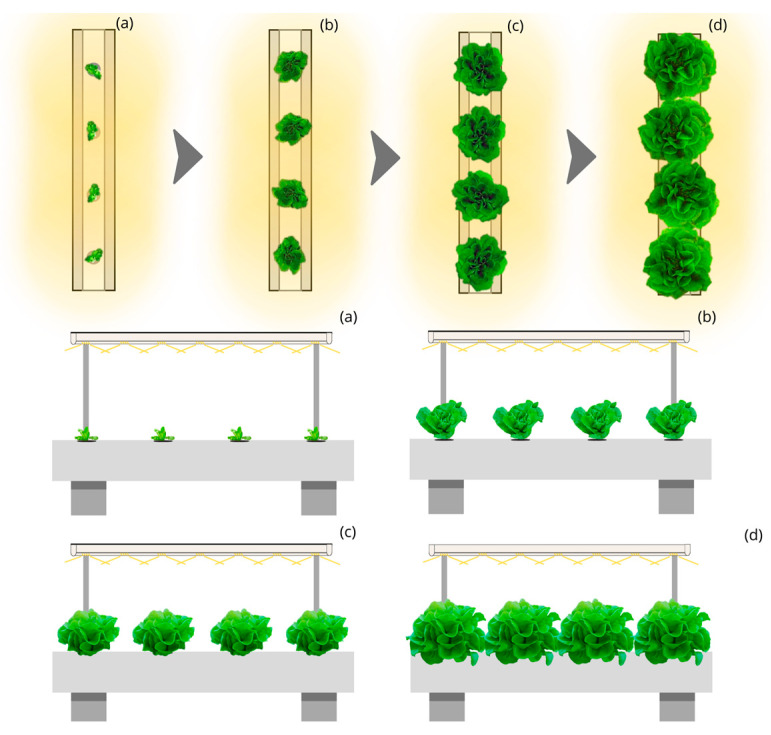
Top and side view of the simulated indoor farming system under the Linear static light and Static Culture Bed with Linear Planting (LSL-LPS) design for four stages (**a**–**d**) of lettuce growth.

**Figure 2 plants-12-03456-f002:**
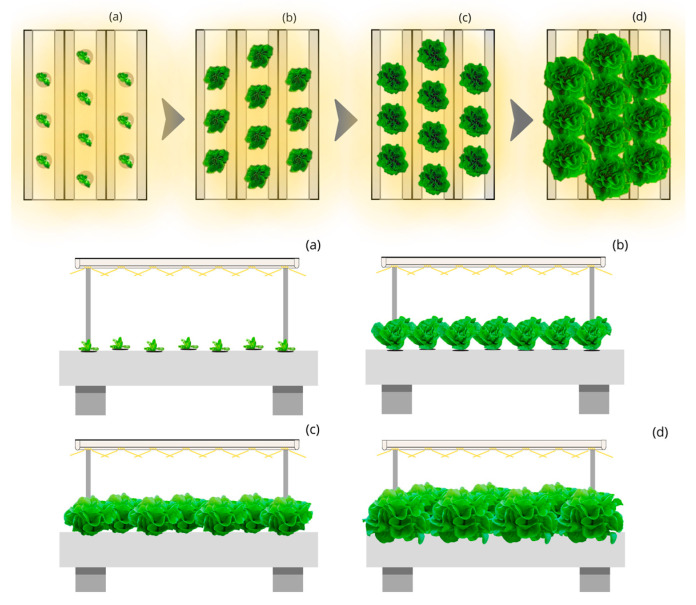
Top and side view of the simulated indoor farming system under the Linear static light and Static culture bed with quincunx planting (LSL-QS) design for four stages (**a**–**d**) of lettuce growth.

**Figure 3 plants-12-03456-f003:**
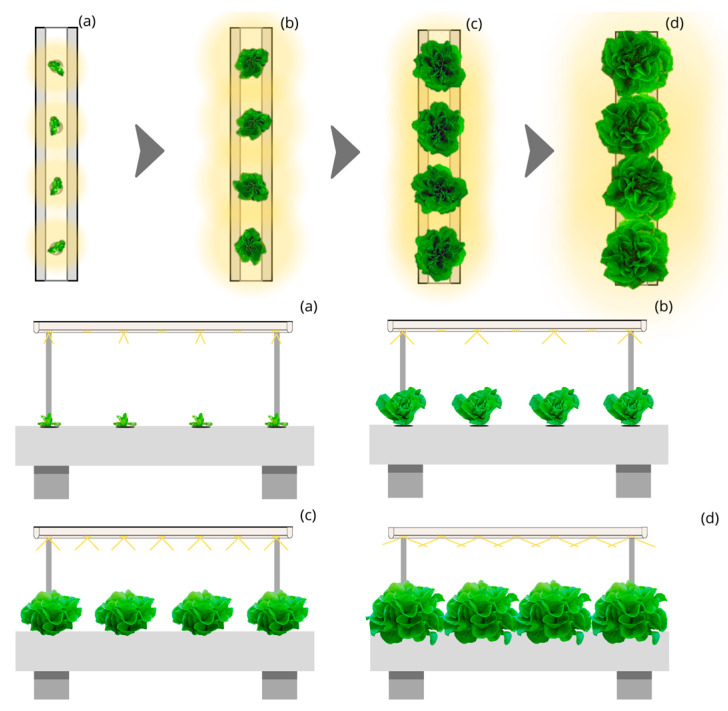
Top and side view of the simulated indoor farming system under the Circular Moving Light and Static Culture Bed with Linear Planting (CML-LPS) design for four stages (**a**–**d**) of lettuce growth.

**Figure 4 plants-12-03456-f004:**
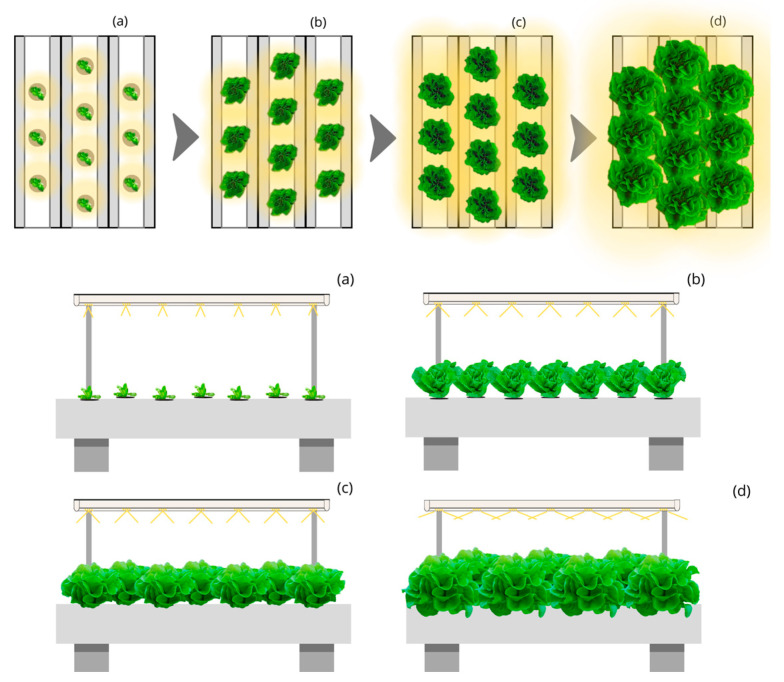
Top and side view of the simulated indoor farming system under the Circular Moving Light and Static Culture Bed with Quincunx Planting (CML-QS) design for four stages (**a**–**d**) of lettuce growth.

**Figure 5 plants-12-03456-f005:**
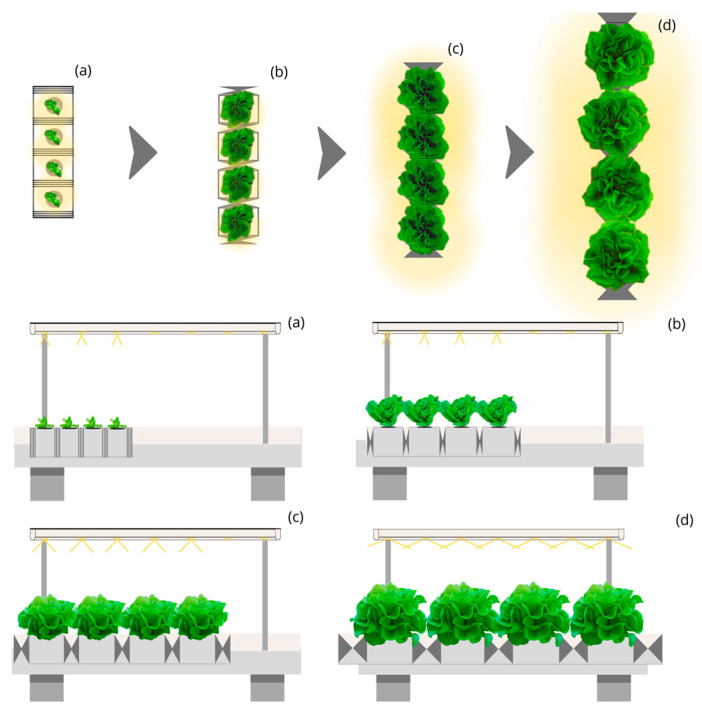
Top and side view of the simulated indoor farming system under the Circular Moving Light and Mobile Culture Bed with Linear Planting (CML-LPM) design for four stages (**a**–**d**) of lettuce growth.

**Figure 6 plants-12-03456-f006:**
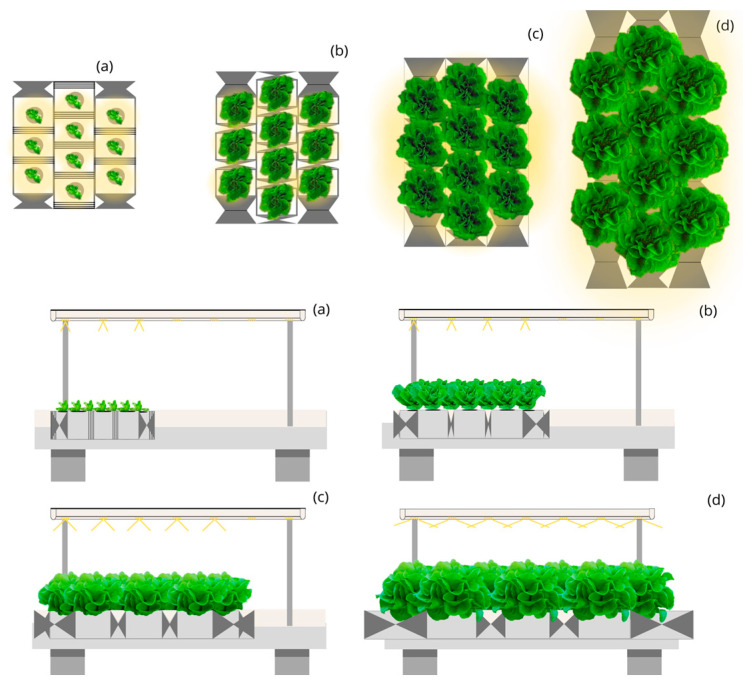
Top and side view of the simulated indoor farming system under the Circular Moving Light and Mobile Culture Bed with Quincunx Planting (CML-QM) design for four stages (**a**–**d**) of lettuce growth.

**Figure 7 plants-12-03456-f007:**
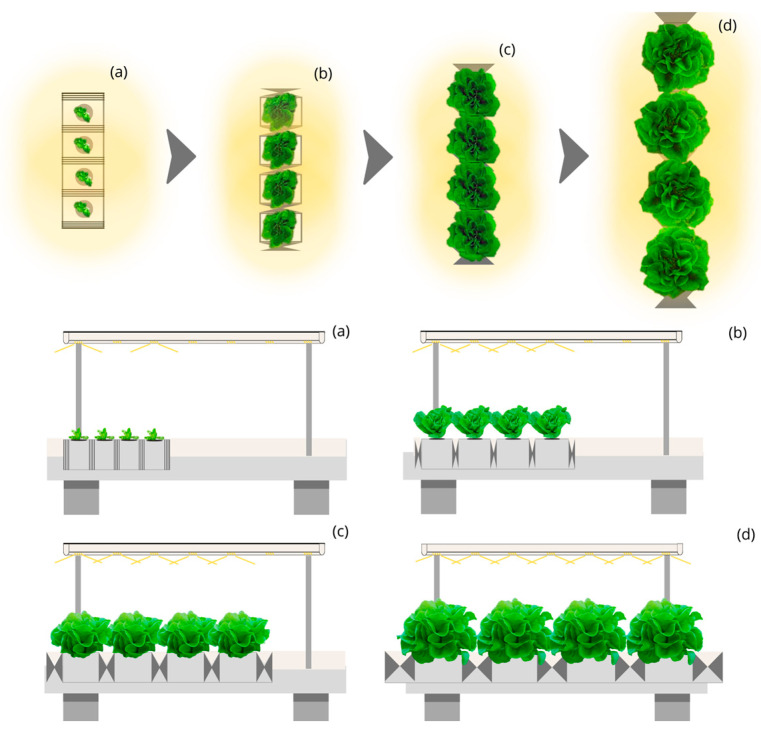
Top and side view of the simulated indoor farming system under the Linear Moving Light and Mobile Culture Bed with Linear Planting (LML-LPM) design for four stages (**a**–**d**) of lettuce growth.

**Figure 8 plants-12-03456-f008:**
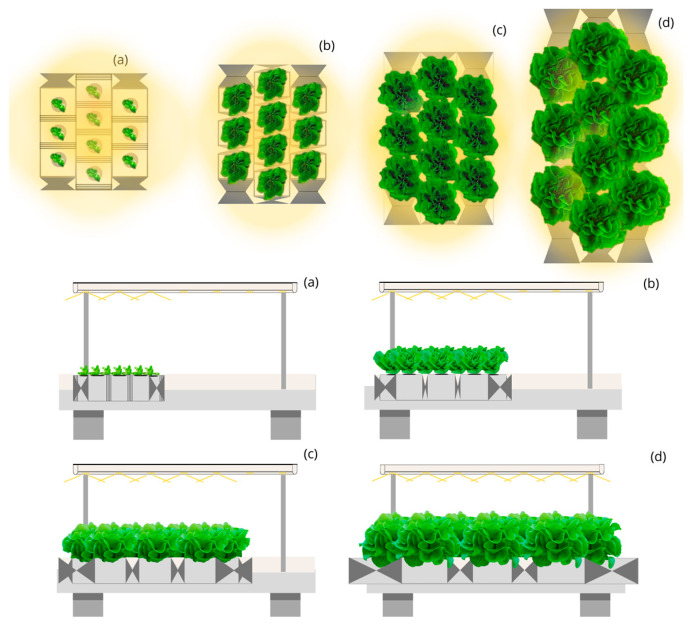
Top and side view of the simulated indoor farming system under the Linear Moving Light and Mobile Culture Bed with Quincunx Planting (LML-QM) design for four stages (**a**–**d**) of lettuce growth.

**Figure 9 plants-12-03456-f009:**
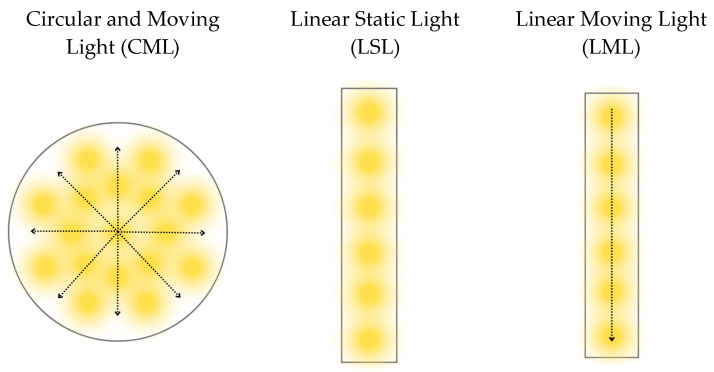
Structural lighting design.

**Figure 10 plants-12-03456-f010:**
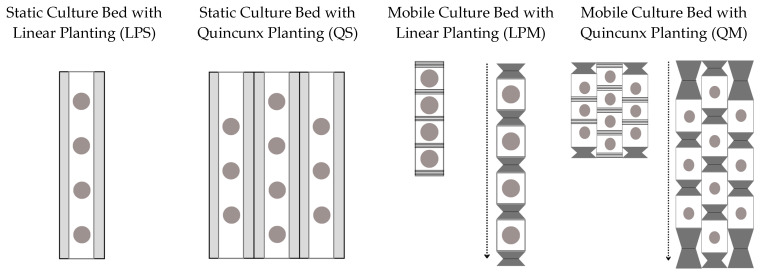
Structural design of cultivation base and planting system.

**Table 1 plants-12-03456-t001:** Simulation of the annual electricity consumption (in thousands of kWh) for producing between 500 and 7500 thousand kilos of lettuce.

	LSL-LPS	LSL-QS	CML-LPS	CML-QS	LML-LPM	LML-QM	CML-LPM	CML-QM	Hypothetical Structures	
	50%	55%	60%	65%	70%	75%	80%	85%	90%	95%
**500**	4152	3775	3460	3194	2966	2768	2595	2442	2307	2185
**1000**	8304	7549	6920	6388	5932	5536	5190	4885	4613	4371
**1500**	12,456	11,324	10,380	9582	8897	8304	7785	7327	6920	6556
**2000**	16,608	15,098	13,840	12,776	11,863	11,072	10,380	9770	9227	8741
**2500**	20,760	18,873	17,300	15,969	14,829	13,840	12,975	12,212	11,533	10,926
**3000**	24,912	22,648	20,760	19,163	17,795	16,608	15,570	14,654	13,840	13,112
**3500**	29,064	26,422	24,220	22,357	20,760	19,376	18,165	17,097	16,147	15,297
**4000**	33,216	30,197	27,680	25,551	23,726	22,144	20,760	19,539	18,454	17,482
**4500**	37,368	33,971	31,140	28,745	26,692	24,912	23,355	21,981	20,760	19,668
**5000**	41,521	37,746	34,600	31,939	29,658	27,680	25,950	24,424	23,067	21,853
**5500**	45,673	41,521	38,060	35,133	32,623	30,448	28,545	26,866	25,374	24,038
**6000**	49,825	45,295	41,521	38,327	35,589	33,216	31,140	29,309	27,680	26,223
**6500**	53,977	49,070	44,981	41,521	38,555	35,984	33,735	31,751	29,987	28,409
**7000**	58,129	52,844	48,441	44,714	41,521	38,752	36,330	34,193	32,294	30,594
**7500**	62,281	56,619	51,901	47,908	44,486	41,521	38,925	36,636	34,600	32,779

First row: Simulated and hypothetical structures; second row: Percentage of photon capture; first column: Lettuce annual production in thousand kilos.

**Table 2 plants-12-03456-t002:** Simulation of the annual electrical energy expenses (in thousands of EUR) for producing between 500 and 7500 thousand kilos of lettuce.

	LSL-LPS	LSL-QS	CML-LPS	CML-QS	LML-LPM	LML-QM	CML-LPM	CML-QM	Hypothetical Structures	
	50%	55%	60%	65%	70%	75%	80%	85%	90%	95%
**500**	955	868	796	735	682	637	597	562	531	503
**1000**	1910	1736	1592	1469	1364	1273	1194	1123	1061	1005
**1500**	2865	2604	2387	2204	2046	1910	1791	1685	1592	1508
**2000**	3820	3473	3183	2938	2728	2547	2387	2247	2122	2010
**2500**	4775	4341	3979	3673	3411	3183	2984	2809	2653	2513
**3000**	5730	5209	4775	4408	4093	3820	3581	3370	3183	3016
**3500**	6685	6077	5571	5142	4775	4457	4178	3932	3714	3518
**4000**	7640	6945	6366	5877	5457	5093	4775	4494	4244	4021
**4500**	8595	7813	7162	6611	6139	5730	5372	5056	4775	4524
**5000**	9550	8682	7958	7346	6821	6366	5969	5617	5305	5026
**5500**	10,505	9550	8754	8081	7503	7003	6565	6179	5836	5529
**6000**	11,460	10,418	9550	8815	8185	7640	7162	6741	6366	6031
**6500**	12,415	11,286	10,346	9550	8868	8276	7759	7303	6897	6534
**7000**	13,370	12,154	11,141	10,284	9550	8913	8356	7864	7428	7037
**7500**	14,325	13,022	11,937	11,019	10,232	9550	8953	8426	7958	7539

First row: Simulated and hypothetical structures; second row: Percentage of photon capture; first column: Lettuce annual production in thousand kilos.

## Data Availability

Not applicable.

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
