# Peer review of "Light Energy Efficiency in Lettuce Crop: Structural Indoor Designs Simulation"

_plants, 2023, doi:10.3390/plants12193456_

Round 1

Reviewer 1 Report

The MS "The efficiency of light energy applied to indoor agriculture  structures" is well-written and structured. The aim is clear and results are properly presented and discussed. 

1. I would suggest to consider the possible change of the title. In the present form it is much broader than the MS content.  

2. It is mentioned that the calculations of the cost of production of indoor lettuce did not accounted additional costs of production, such as the packaging and distribution of the product. What about initial costs? What is approximate cost differnce between culture systems and light desings used in the study?  

Author Response

Dear Reviewer

We appreciate your time and the comments made in the manuscript. Following are our responses to your two points:

Point 1. I would suggest to consider the possible change of the title. In the present form it is much broader than the MS content.  

Response 1: We changed the title to "Light Energy Efficiency in Lettuce Crop: Structures Indoor Designs Simulation" according to the aim objective and the obtained results.

Point 2. It is mentioned that the calculations of the cost of production of indoor lettuce did not accounted additional costs of production, such as the packaging and distribution of the product. What about initial costs? What is approximate cost differnce between culture systems and light desings used in the study?  

Response 2: Considering the research's specific objective, which focuses on optimizing the photonic capture received by lettuce during its growth, thereby achieving economic savings in the electricity bill attributed exclusively to the distribution of luminaires, initial investments are not within the scope of this investigation.

Our attention is exclusively directed toward LED lamps, given that they constitute the predominant share of energy expenditure in indoor agriculture, as corroborated by the citations provided below. Including supplementary costs would necessitate a more comprehensive and detailed cost-benefit analysis comparing conventional and indoor cultivation.

  1. Bugbee, B. Economics of LED lighting. In Light emitting diodes for agriculture; Gupta, S.D. Springer, 2017; Chapter 5, pp. 81-99. ISBN: 978-981-10-5806-6. doi: 10.1007/978-981-10-5807-3
  2. Van Iersel, M.W. Optimizing LED lighting in controlled environment agriculture. In Light emitting diodes for agriculture; Gupta, S.D. Springer, 2017; Chapter 4, pp. 59-80. ISBN: 978-981-10-5806-6. doi: 10.1007/978-981-10-5807-3

Once again, thank you very much.

Reviewer 2 Report

The manuscript considers the analysis of possible layouts of the grops and lighting in indoor agriculture. On this basis Authors try to analyze the potentially most efficient layouts. I believe the problem is important and should be deepend, but in my opinion this manuscript needs additional work. For this reason I suggest rejecting the manuscript to give time to the Authors to improve the content. 

My main comments:

1. The organization of the manuscript is suprising - Materials and Methods paragraph is always going before Results and Discussion

2. Some calculations/simmulations were performed - it is not clear if any and what software was used.

3. I haven't found any assumptions, simpliciations mentioned in the text and I think there are many of them. For example I have serious doubts considering Figure 9 - how did you perform the design of lighting? How do you know what will be the light distribution? What was the photometry of the lamps you used and the height? Was lighitng optimized or were these layouts just various concepts? Are you sure you obtain correct levels of PPFD? Do you have the information baout the uniformity of photon flux on the surface of the crop? There are no basic information to evaluate lighting system. Correct design of illumination is fundamental when energy efficiency is analyzed. 

4. It seems you forgot about energy consumption for movable/mobile structures - only some of cases are non-static and you forget that engines also consume electricity which should be included in the analysis. 

Author Response

Dear Reviewer

We appreciate your time and the comments made in the manuscript. Following there are our responses to your points:

Point 1. The organization of the manuscript is suprising - Materials and Methods paragraph is always going before Results and Discussion.

Response 1: Plants have published the template with the item order we presented in the manuscript. Plants | Instructions for Authors (mdpi.com)

Point 2. Some calculations/simmulations were performed - it is not clear if any and what software was used.

Response 2: Following your comment, we added in item 4.1. Variables simulation the use of Microsoft Excel 2016 spreadsheet.

Point 3. I haven't found any assumptions, simpliciations mentioned in the text and I think there are many of them. For example I have serious doubts considering Figure 9 - how did you perform the design of lighting? How do you know what will be the light distribution? What was the photometry of the lamps you used and the height? Was lighitng optimized or were these layouts just various concepts? Are you sure you obtain correct levels of PPFD? Do you have the information baout the uniformity of photon flux on the surface of the crop? There are no basic information to evaluate lighting system. Correct design of illumination is fundamental when energy efficiency is analyzed. 

Response 3: This is an initial stride towards characterizing the electrical consumption while proposing strategies for cost optimization based on the proportion (percentage) of photon capture on leaf surfaces, contingent upon the distribution pattern of luminaires and culture bed. The methodology (references below) does not consider quantifying Photosynthetic Photon Flux Density (PPFD); consequently, parameters such as the uniformity of photon flux or the photometry of the lamps are not considered.

As a non-experimental investigation, the design of lighting and their distribution inspiration from diverse horticultural illumination technologies developed by Koninklijke Philips N.V. company. This includes the assessment of photon output (luminous flux) and electrical input (power consumption) to deduce the photonic efficacy of the light source. The authors have added such information in lines 289 and 362 of the manuscript.

  1. Bugbee, B. Turning photons into food. Calculating potential yield in optimal environments. Utah State University, Utah, United States of America, 20/10/2019. Available online: https://www.youtube.com/watch?v=wsaufB5F8dk&t=146s
  2. Volk, T.; Bugbee, B.; Wheeler, R. An approach to crop modeling with the energy cascade. Life support & biosphere science: international journal of earth space 1995, 119-27.
  3. Bugbee, B.G.; Salisbury, F.B. Exploring the limits of crop productivity. I. Photosynthetic efficiency of wheat in high irradiance environments. Plant Physiol. 1988; 88(3):869-78. doi: 10.1104/pp.88.3.869.

Point 4. It seems you forgot about energy consumption for movable/mobile structures - only some of cases are non-static and you forget that engines also consume electricity which should be included in the analysis. 

Response 4: Considering the research's specific objective, which focuses on optimizing the photonic capture, and thereby achieving economic savings in the electricity bill attributed exclusively to the distribution of luminaires and culture bed; initial investment or the calculation of additional equipment such as controllers and automation systems are not within the scope of this investigation.

Our attention is exclusively directed toward LED lamps, given that they constitute the predominant share of energy expenditure in indoor agriculture, as corroborated by the citations provided below. Including supplementary costs would necessitate a more comprehensive and detailed cost-benefit analysis.

  1. Buggbe, B. Economics of LED lighting. In Light emitting diodes for agriculture; Gupta, S.D. Springer, 2017; Chapter 5, pp. 81-99. ISBN: 978-981-10-5806-6. doi: 10.1007/978-981-10-5807-3
  2. Van Iersel, M.W. Optimizing LED lighting in controlled environment agriculture. In Light emitting diodes for agriculture; Gupta, S.D. Springer, 2017; Chapter 4, pp. 59-80. ISBN: 978-981-10-5806-6. doi: 10.1007/978-981-10-5807-3

Once again, thank you very much.

Reviewer 3 Report

Dear Authors,

The study is the interesting and worth to publish. However, I have few comments:

In Introduction, the previous studies (if any) is worth to mention;

In M&M, it is not clear how many times the experiment was repeated.

Minor editing of English language required.

Author Response

Dear Reviewer

We appreciate your time and the comments made in the manuscript. Following there are our responses to your points:

Point 1. In Introduction, the previous studies (if any) is worth to mention.

Response 1: In the literature, no previous investigations focused on optimizing the photonic uptake that plants receive through the distribution of luminaires and growing beds. However, in line 65 three references (13-15) are cited, which are investigations that analyze and propose mobile structures in indoor systems for the transport of plant material, demonstrating the opportunity and need to build and develop mobile models during the crop's growth stage until harvest.

Point 2. In M&M, it is not clear how many times the experiment was repeated.

Response 2: This is a non-experimental investigation based on simulating eight indoor structures. The need for repetitions was obviated due to the distinct conditions exhibited by each structure, directly influencing the variable "Percentage of Photon Capture" (Tables 1 and 2). Furthermore, the study centered on the simulation of a singular crop (lettuce), where the variables within subsection "4.1.1. Yield in Dry Biomass by Photosynthetic Photons" were treated as unique conditions, rendering repetition inconsequential. Ultimately, this research is a foundational step in characterizing electrical consumption and optimizing photon capture by distributing luminaires and culture beds. These hypothetical scenarios provide an initial framework for future investigation, incorporating additional variables or comparative analyses.

Once again, thank you very much.

Round 2

Reviewer 2 Report

Dear Authors,

Thank you for the clarifications, also included in the response to other reviewers. I believe with the new title the manuscript sounds better, although I do not agree with neglecting the energy demand for moving the system. Point 4.1.4. is entitled "Crop costs" -  I believe avoiding some costs is wrong in such an approach - Please discuss that and clarify the limitations of your research. There are many of them. One of the limitations is assuming specific products with specific parameters. In my opinion, at least the curves for the spatial distribution of light of the assumed luminaires should be included together with the technical data of the luminaires (e.g. the photon flux) - giving information about the manufacturer is not necessary in such case.

Author Response

Dear Reviewer,   Thank you for your valuable feedback on our manuscript. We have carefully considered your comments, and we would like to address the points you raised.   Comments: Thank you for the clarifications, also included in the response to other reviewers. I believe with the new title the manuscript sounds better, although I do not agree with neglecting the energy demand for moving the system. Point 4.1.4. is entitled "Crop costs" -  I believe avoiding some costs is wrong in such an approach - Please discuss that and clarify the limitations of your research. There are many of them. One of the limitations is assuming specific products with specific parameters. In my opinion, at least the curves for the spatial distribution of light of the assumed luminaires should be included together with the technical data of the luminaires (e.g. the photon flux) - giving information about the manufacturer is not necessary in such case.   Response: Regarding your observation about the title "4.1.4. Crop costs," you are absolutely correct. This section indeed focuses on the electrical energy consumed for lighting and its annual cost, as well as the cost per kilogram produced. To avoid any confusion, we have revised the section's title to "Cost of Electrical Energy Consumed for Lighting."   Regarding the need to clarify the limitations of our study, we added to the discussion the description of some and an analysis of possible avenues for future research. As you correctly pointed out, simulations rely on assumptions to approximate reality, and we have made an effort to highlight these assumptions in the discussion. We agree that future work should aim to incorporate additional variables to reduce these assumptions and develop more realistic and experimental case studies that represent opportunities to improve horticultural design.   Finally, the designs presented in the manuscript do not represent real lamps, we include the Philips reference as an ideal technological model for indoor horticulture in general. To avoid possible confusion, we have decided to remove this reference. Since these designs are not real lamps, photometric curves will not be incorporated, but we will provide photon flux and input electrical energy information in 4.2.1. Illumination.
. Once again, thank you for your comments.   Sincerely, Luisa F. Lozano Castelllanos    

Round 3

Reviewer 2 Report

Thank you for clarifications in the manuscript.